# Vitamin D Prevents High Glucose-Induced Lipid Droplets Accumulation in Cultured Endothelial Cells: The Role of Thioredoxin Interacting Protein

**DOI:** 10.3390/biomedicines9121874

**Published:** 2021-12-10

**Authors:** Roberta Scrimieri, Alessandra Cazzaniga, Sara Castiglioni, Jeanette A. M. Maier

**Affiliations:** 1Department of Biomedical and Clinical Sciences “Luigi Sacco”, Università di Milano, 20157 Milano, Italy; alessandra.cazzaniga@unimi.it (A.C.); sara.castiglioni@unimi.it (S.C.); jeanette.maier@unimi.it (J.A.M.M.); 2Interdisciplinary Centre for Nanostructured Materials and Interfaces (CIMaINa), Università di Milano, 20133 Milano, Italy

**Keywords:** d-glucose, endothelial cell, vitamin D, oxidative stress, lipid metabolism

## Abstract

Vitamin D (VitD) exerts protective effects on the endothelium, which is fundamental for vascular integrity, partly by inhibiting free radical formation. We found that VitD prevents high glucose-induced Thioredoxin Interacting Protein (TXNIP) upregulation. Increased amounts of TXNIP are responsible for the accumulation of reactive oxygen species and, as a consequence, of lipid droplets. This is associated with increased amounts of triglycerides as the result of increased lipogenesis and reduced fatty acid oxidation. Remarkably, VitD rebalances the redox equilibrium, restores normal lipid content, and prevents the accumulation of lipid droplets. Our results highlight TXNIP as one of the targets of VitD in high glucose-cultured endothelial cells and shed some light on the protective effect of VitD on the endothelium.

## 1. Introduction

Vascular endothelial cells (EC) form a quiescent monolayer that coats the inner lumen of all vessels and retains critical functions that are essential to preserve the integrity of the vasculature and, consequently, health [1]. They act as a metabolic interface between the blood and tissues and ensure optimal nutrient and oxygen delivery to all of the tissues [2]. EC are steadily exposed to glucose, which is taken up from the blood, mainly through the glucose transporter 1 (GLUT1). Glucose is partially utilized for endothelial metabolic needs and is also delivered to the surrounding cells and tissues. Rather than shunting glucose to oxidative phosphorylation to maximize adenosine triphosphate (ATP) production, EC rely on glycolysis, which takes place in the cytosol and does not demand oxygen [3]. As a consequence, oxygen is saved to be delivered to the parenchymal tissues, protecting the EC against the accumulation of Reactive Oxygen Species (ROS), which are typically produced during oxidative phosphorylation [4]. Additionally, the high extent of Fatty Acid β-Oxidation (FAO) in quiescent EC makes important contributions to the maintenance of the redox balance [5]. Therefore, EC sustain different metabolic pathways to protect themselves against oxidative stress, which is the root of endothelial dysfunction [6]. It is avowed that high glucose levels are detrimental for the endothelium both in the large vessels as well as in the microvasculature [2]. When fasting, the EC are exposed to about 5 mmol/L (5 mM) d-glucose, a concentration that increases post-prandially and that remains below 7.8 mmol/L (7.8 mM) in healthy people. The failure to contain post-prandial glucose spikes as well as the chronic increase of glucose in uncontrolled diabetic patients ultimately generate endothelial dysfunction [2], partly through the increased endothelial production of free radicals [7,8], and partly through conducted through metabolic reprogramming [9]. High extracellular glucose boosts glucose uptake and metabolism through different pathways, i.e., the polyol pathway, which promotes oxidative stress by consuming Nicotinamide Adenine Dinucleotide Phosphate Hydrogen (NADPH), and glycolytic side branches, such as the hexosamine and the pentose phosphate pathways [2,10]. Moreover, it also results in the activation of Protein-kinase C (PKC) because diacyl glycerol accumulates in response to high intracellular glucose concentrations [11]. As a result of endothelial-altered metabolism caused by high glucose, advanced glycation end products (AGE), which activate the inflammatory response, are generated. Both PKC activation and AGE formation are implicated in the promotion of oxidative stress in EC that have been exposed to high glucose levels [12]. Far less is known about fatty acid metabolism in EC. FAO plays an important role in endothelial homeostasis both in vitro and in vivo [13]. It sustains nucleotide synthesis, fuels the tricarboxylic acid cycle and, as mentioned above, contributes to redox homeostasis via the synthesis of NADPH [14]. A particular light has been shed on Carnitine Palmitoyltransferase 1A (CPT1A), a crucial enzyme that converts long-chain acyl-CoAs into long-chain acyl-carnitines, which then enter the mitochondria where FAO takes place [15]. Silencing *CPT1A* in the EC triggers oxidative stress and the overexpression of genes that are involved in controlling redox balance [14].

Remarkably, EC can generate lipid droplets and dynamic cytosolic fat storage compartments [16]. This means that EC can store neutral lipids, which then provide fatty acids to be metabolized in the mitochondria or to be transported to nearby tissues [16]. It is also emerging that lipid droplets are critical components of the cellular stress response, as they protect against lipotoxicity [16]. Lipid supplementation for 24h to cultured EC results in the reversible accumulation of lipid droplets, and similar results were observed in the aortic endothelium of hypertriglyceridemic mice [17].

Beyond its essential role in bone health, Vitamin D (VitD) exerts protective effects on the endothelium. Indeed, VitD deficiency is related to endothelial dysfunction, partially because of the downregulation of the VitD Receptor (VDR) [18]. Consistently, VitD supplementation in VitD-deficient diabetic patients improved endothelial function [19], and a recent systematic review and metanalysis of randomized clinical trials demonstrated that VitD supplementation decreases circulating inflammatory cytokines in patients with altered glucose tolerance [20]. However, another study reported no significant effects of VitD supplementation on endothelial dysfunction [21]. The results that have been obtained in vitro sustain a beneficial effect of VitD in the EC. Indeed, in Human Umbilical Vein Endothelial Cells (HUVEC), 1,25(OH)_2_D_3_ (calcitriol), the most active metabolite of VitD, prevents leptin-induced endothelial dysfunction in a VDR-dependent fashion [22]. Moreover, in HUVEC treated with acetoacetate in order to mimic ketosis, VitD inhibits ROS formation and monocyte adhesion [19]. Our study sought to address some fundamental questions on the response of HUVEC to high d-glucose and on the potential protective role of VitD. Initially, we investigated the levels of some pro- and antioxidant proteins. Then, we studied the contribution of oxidative stress in reprogramming lipid metabolism. Finally, we focused on the effects of VitD in protecting HUVEC from oxidative stress, metabolic derangements, and lipid droplet accumulation.

## 2. Materials and Methods

### 2.1. Cell Culture

HUVEC were purchased from the American Type Culture Collection (ATCC, Manassas, WV, USA), cultured in medium M199 (Euroclone, Milano, Italy) containing 10% Fetal Bovine Serum (FBS) (Euroclone), 1 mM l-Glutamine (Euroclone), 1 mM Sodium Pyruvate (Sigma-Aldrich, St. Louis, MO, USA), 1 mM Penicillin-Streptomycin (Euroclone), 5 U/mL Heparin (Sigma-Aldrich), and 150 µg/mL Endothelial Cell Growth Supplement (Sigma-Aldrich) on collagen-coated dishes (50 µg/mL) (Sigma-Aldrich). The cells were routinely tested for the expression of endothelial markers and were used for 6–7 passages. d-glucose (Sigma-Aldrich) was used at the concentrations of 11.1 mM and 30 mM, and l-glucose (Sigma-Aldrich) was used as a control of osmolarity at the concentration of 30 mM. After testing 1α,25-Dihydroxyvitamin D_3_ (VitD) (Sigma-Aldrich) cytotoxicity in a dose-dependent fashion by 3-(4,5-dimethylthiazol-2-yl)-2,5-diphenyl-2H-tetrazolium bromide (MTT) assay (the data are available at the following link https://dataverse.unimi.it/dataverse/biomedicines/ accessed on 30 September 2021), VitD was used at the concentration of 20 nM. Thioredoxin Interacting Protein (TXNIP) was inhibited using small interfering RNAs (siRNAs). Subconfluent cells were transfected with siRNAs targeting *TXNIP* (20 nmol, 5′-AAGCCGTTAGGATCCTGGCT-3′ (Qiagen, Hilden, Germany)). All the non-silenced samples were transfected with a scrambled non-silencing sequence (NS) (20 nmol, Qiagen, cat. n° 1027310) that was used as a control, which produced the same results as the non-transfected samples (data not shown). Lipofectamine RNAiMAX was used as a transfection reagent (Invitrogen, Carlsbad, CA, USA) and was used according to the manufacturer’s recommendations. After 6h, the siRNA transfection medium was replaced with a culture medium with the addition of either 11.1 mM or 30 mM of d-glucose in the presence or not of 20 nM of VitD. In some experiments, N-acetylcysteine (NAC, 5 mM) (Sigma-Aldrich) was used as antioxidant [23], in combination or not with high glucose. All the experiments were performed in triplicate.

### 2.2. ROS Production

For ROS detection, confluent HUVEC were cultured in a 96-well black plate (Greiner Bio-One, Kremsmunster, Austria) and, at the end of the experiments, they were incubated for 30 min with 10 mM 2′-7′-dichlorofluorescein diacetate (DCFH) solution (Sigma-Aldrich). The DCFH dye emission was monitored at 535 nm (excitation λ = 484 nm) using the Varioskan LUX Multimode Microplate Reader (Thermo Fisher Scientific, Waltham, MA, USA). Then, the cells were fixed in Phosphate Buffered Saline (PBS) containing 3% paraformaldehyde (PFA) and 2% sucrose (pH 7.6) for 30 min and, after extensive washing, the cells were incubated with 4′,6-diamidino-2-phenylindole (DAPI), which was used to stain the nuclei (1:10,000). DAPI florescence (λ_ex/em_ = 350/470 nm) was monitored using the Varioskan LUX Multimode Microplate Reader (Thermo Fisher Scientific) and was used to normalize the DCFH dye emission [7]. The results are the mean of three independent experiments performed in triplicate ± SD.

### 2.3. Triglyceride Quantification

Triglycerides were quantified using a Triglyceride Quantification Kit (Sigma-Aldrich) according to the manufacturer’s recommendations. Briefly, triglycerides are broken down into free fatty acids and glycerol, which is then oxidized to generate a fluorescent product (λ_ex/em_ = 535/587 nm). Fluorescence was monitored using the Varioskan LUX Multimode Microplate Reader (Thermo Fisher Scientific). Prior to the extraction of the triglycerides, the cells were trypsinized. An aliquot was stained with 0.4% trypan blue solution and were counted using a Luna Automated Cell Counter (Logos Biosystems, Anyang-si, Gyeonggi-do, Korea). The fluorescent results were normalized on the cell number. The results are the mean of three independent experiments that were performed in triplicate ± SD.

### 2.4. Staining of Neutral Lipids

HUVEC were seeded to detect lipids as well as to perform the MTT assay under the same experimental conditions. Oil Red O Staining was used to detect neutral lipids. After the 24h treatments, the cells were washed three times with PBS, fixed in PFA 10% for 30 min at room temperature, washed once again with PBS, and then stained with 60% filtered Oil Red O stock solution (Sigma-Aldrich) for 20 min. After extensive washing, the Oil Red O was solubilized in 100% isopropanol, was quantified by measuring the absorbance at 500 nm, and was normalized to the cell number by MTT assay [24,25] after image acquisition using FLoid Cell Imaging Station (Thermo Fisher Scientific). To further confirm the results, staining with BODIPY 493/503 was performed (see dataverse at the following link https://dataverse.unimi.it/dataverse/biomedicines/ accessed on 30 September 2021). The results are the mean of three independent experiments performed in triplicate ± SD.

### 2.5. Western Blot Analysis

HUVEC were lysed in 50 mM Tris-HCl (pH 7.4) containing 150 mM NaCl (Sigma-Aldrich), 1% NP40, 0.25% sodium deoxycholate (Sigma-Aldrich), protease inhibitors (10 µg/mL Leupeptin, 10 µg/mL Aprotinin and 1 mM Phenylmethylsulfonyl fluoride, PMSF) (Sigma-Aldrich), and phosphatase inhibitors (1 mM sodium fluoride, 1 mM sodium vanadate, 5 mM sodium phosphate) (Sigma-Aldrich). Lysates (40 µg/lane) were separated on SDS–PAGE and were transferred to nitrocellulose sheets using the Trans-Blot Turbo Transfer System (Biorad, Hercules, CA, USA.). Western Blot analysis was performed using antibodies against TXNIP (Thermo Fisher Scientific, cat. n° 40–3700, 1:250), Sirtuin 1 (SIRT1) (Thermo Fisher Scientific, cat. n° PA5–17074, 1:1000), Sirtuin 2 (SIRT2) (Millipore, Vimodrone, Italy, cat. n° 09–843, 1:4000), Superoxide-dismutase 2 (SOD2) (BD Transduction Laboratories, Milano, Italy, cat. n° 611580, 1:1000), Heat Shock Protein 70 kilodaltons (HSP70) (Santa Cruz Biotechnology, Dallas, TX, USA, cat. n° sc-1060, 1:200), EDF1 (Aviva Biosciences, San Diego, CA, USA, cat. n° ARP37729_T100, 1:500), PPARɣ (Santa Cruz, cat. n° sc-7196, 1:200), and CPT1A (Thermo Fisher Scientific, cat. n° 15184-1-AP, 1:1000). Actin (Santa Cruz, cat. n° sc-1616, 1:200) was used as the equal loading control. After extensive washing, secondary antibodies labelled with horseradish peroxidase (GE Healthcare, Waukesha, WI, USA) were used. Immunoreactive proteins were detected by the SuperSignal Chemiluminescence Kit (Thermo Fisher Scientific). A representative blot is shown. The densitometric analysis was performed using Image J Lab software (Biorad). The results are the mean of three independent experiments ± SD.

### 2.6. Fatty Acid Oxidation

FAO, the primary metabolic pathway for the degradation of fatty acids, was monitored by Fatty Acid Oxidation assay (Abcam, Cambridge, UK) in living cells. The cells were seeded in a 96-well black plate (Greiner Bio-One), and upon treatment with high glucose for 24 h, they were rinsed twice with pre-warmed Fatty Acid-Free medium followed by the addition of pre-warmed Fatty Acid Measurement Medium. Extracellular O_2_ Consumption Reagent (Abcam) was added into all the wells except for the blank control well. The FAO activator carbonyl cyanide-p-trifluoromethoxyphenylhydrazone (FCCP, 0.625 μM) was used as the positive control. Finally, the wells were sealed with pre-warmed high-sensitivity mineral oil. Subsequently, the 96-well black plate was placed into the Varioskan LUX Multimode Microplate Reader (Thermo Fisher Scientific), which had been pre-set to 37 °C. The fluorescent signal (λ_ex/em_ = 380/650 nm) was measured every 2 min for 180 min. To simplify the procedure, the results were expressed in a box plot graph. The results are the mean of three independent experiments performed in triplicate ± SD.

### 2.7. Statistical Analysis

Data are reported as means ± standard deviation (SD). The data were normally distributed, and they were analyzed using the two-way repeated measures ANOVA. The *p*-values deriving from multiple pairwise comparisons were corrected using the Bonferroni method. Statistical significance was defined for *p*-value < 0.05. Concerning the figures, * *p* < 0.05; ** *p* < 0.01; *** *p* < 0.001; **** *p* < 0.0001.

## 3. Results

### 3.1. TXNIP Is Upregulated in HUVEC Exposed to High Glucose

To get insights into the mechanisms that are involved in high glucose-triggered oxidative stress, we analysed the levels of some of the proteins that are implicated in the control of redox balance. HUVEC were cultured in media containing physiological (5.5 mM, CTR) or high glucose (11.1 mM and 30 mM) concentrations for 24 h. l-glucose (30 mM) was used as a control for osmolarity. As shown in Figure 1, we found a substantial increase in the total amounts of TXNIP. When the cells were cultured at the highest d-glucose concentration (30 mM), we also observed the significant downregulation of SIRT1, the most evolutionarily conserved member of the sirtuin family, which exerts beneficial effects on the endothelium. HSP70, PON2, SIRT2, and SOD2 were not modulated. l-glucose exerted no effects (Figure 1).

### 3.2. TXNIP Upregulation Accounts for ROS Accumulation in HUVEC Cultured in High Glucose

In this paper, we focused on the role of TXNIP upregulation in high glucose-treated HUVEC. For this purpose, we transiently silenced the cells with specific siRNAs targeting *TXNIP*. Then, HUVEC were cultured for 24 h in medium containing physiological (5.5 mM, CTR), high-extracellular d-glucose (11.1 mM and 30 mM) or l-glucose (30 mM) as a control. Figure 2A shows that *TXNIP* silencing prevents high glucose-induced TXNIP increase. Moreover, upon *TXNIP* silencing, high glucose-induced ROS production was dampened by the same amount after the administration of the antioxidant NAC (5 mM), the precursor of glutathione that is widely used as an antioxidant (Figure 2B).

### 3.3. VitD Prevents TXNIP Upregulation in HUVEC Cultured in High Glucose

Since (i) TXNIP was initially characterized as a target of VitD and since (ii) VitD exerts a protective effect upon metabolic challenge in HUVEC, we anticipated that VitD might affect the levels of TXNIP found in HUVEC cultured in high glucose conditions. Therefore, we exposed HUVEC to media containing high levels of glucose in the presence or in the absence of VitD (20 nM) for 24 h. VitD counters high glucose-induced TXNIP upregulation and ROS accumulation (Figure 3A,B).

### 3.4. VitD Hinders Lipid Droplets Formation in HUVEC Exposed to High Glucose

In several cell types, the accumulation of lipid droplets storing triglycerides is interpreted as an adaptive response to stress. To test whether HUVEC behave in a similar manner, we initially evaluated the amounts of triglycerides in HUVEC that had been cultured in media containing high levels of glucose. Control and l-glucose-cultured cells contain a certain amount of triglycerides that is dose-dependently increased upon exposure to 11.1 mM and 30 mM of d-glucose for 24 h. Moreover, the silencing of *TXNIP* as well as the treatment with VitD reduced the triglyceride amounts to normal physiological levels (Figure 4A). We then stained HUVEC, exposed to high levels of glucose for 24 h, with Oil Red O to detect neutral lipids to analyse the potential role of TXNIP in driving the accumulation of lipids. Moreover, since treatment with VitD downregulates TXNIP, we also treated the cells with VitD. It is noteworthy that, at baseline, HUVEC contain lipid droplets and that high d-glucose levels increase their number. Interestingly, the high glucose-induced deposition of lipids is dampened by *TXNIP* silencing as well as by the addition of VitD (Figure 4B).

### 3.5. VitD Corrects High Glucose-Induced Imbalance of Lipid Metabolism in HUVEC

To shed some light on the pathways leading to the deposition of triglycerides, we analysed some key markers that have been found to be involved in lipid metabolism. First, we focused our attention on some molecules that are involved in lipogenesis. We analysed the modulation of PPARγ and its transcriptional coactivator EDF1, both of which are required for lipogenesis. In high glucose level-cultured cells, Western blot revealed a significant upregulation of both EDF1 and PPARγ, which was averted by VitD (Figure 5A) and *TXNIP* silencing (Figure 5B). Secondly, we analysed the expression of CPT1A, an enzyme that is located on the mitochondrial membrane and that is involved in the transport of fatty acids into the mitochondria to undergo β-oxidation. The total amount of CPT1A was reduced in the cells that had been cultured in high glucose-containing media, and VitD rescued it to normal levels, as did *TXNIP* silencing (Figure 5A,B). This result is in accordance with the decreased β-oxidation rate that was measured in the high glucose-cultured cells, which was recovered in the presence of VitD and after *TXNIP* silencing (Figure 5D).

## 4. Discussion

Observational data have consistently established low serum concentrations of VitD in patients with Type 2 Diabetes Mellitus (T2D). Of note, it seems that the duration of diabetes rather than glycemic control is associated with VitD deficiency [26]. Remarkably, observational studies as well as preclinical data support that low VitD correlates with increased risk of hypertension, atherosclerosis, metabolic disorders, and low-grade inflammation, all of which are conditions that share endothelial dysfunction as a common element [27,28,29]. Accordingly, through the high-frequency ultrasonographic imaging of the brachial artery to assess endothelium-dependent flow-mediated vasodilation, it was demonstrated that VitD improves endothelial function in VitD-deficient subjects [30] and in patients with T2D and with low serum levels of VitD [31]. These findings prompted us to investigate the effects of VitD on HUVEC. VitD has been reported to protect HUVEC from hydrogen peroxide-induced oxidative stress by inhibiting superoxide formation as well as to improve antioxidant defenses in HUVEC that have been treated with high concentrations of ketone bodies [19,32]. Here, we show that VitD downregulates TXNIP and, accordingly, mimics the effects of *TXNIP* silencing in HUVEC that have been cultured in high levels of glucose by preventing oxidative stress and by correcting lipid metabolism and storage in lipid droplets.

Despite originally being isolated as a VitD-upregulated protein [33,34], TXNIP is differently regulated by VitD in various cell types [35]. Cell specificity in the modulation of the response to VitD is described, and the downregulation of TXNIP after exposure to VitD seems to occur in cells harboring wild type p53 [35], as HUVEC do. Elevated TXNIP is implicated in the pathogenesis of various complex diseases, including diabetes and neurologic and inflammatory disorders [36]. This is not surprising since TXNIP regulates lipid and glucose metabolism both dependently and independently from the inhibition of thioredoxin (TRX) [36]. Focusing on the endothelium, TXNIP is overexpressed in the vascular EC of many vessels in hypertensive rats and contributes to oxidative stress and endothelial dysfunction in hypertension [7,37,38]. It is upregulated in the aortic endothelium of diabetic rats and in human aortic EC that have been cultured in high levels of glucose, associated with dysfunction in both cases [38]. Interesting results were obtained in endothelial TXNIP knockout mice under metabolic stress since the aorta was protected from damage through antioxidant and anti-inflammatory mechanisms [39]. Additionally, long non-coding RNAs (lnc), which disrupt the stability of the target protein, are involved in the regulation of TXNIP. Indeed, a recent report showed that high glucose-treated HUVEC downregulate lnc-SNHG15, which reduces TXNIP expression by enhancing its ubiquitination [40], thus mitigating high glucose-induced endothelial dysfunction. Our results are in keeping with the increasing evidence pointing to upregulated TXNIP as a player in endothelial dysfunction in response to high glucose levels. Indeed, we found that the downregulation of *TXNIP* by specific siRNAs reduces oxidative stress and the accumulation of triglycerides in lipid droplets in HUVEC.

Lipid droplets are intracellular organelles that store neutral lipids and have been detected in many eukaryotic cell types and have been interpreted as an adaptation mechanism under metabolic stress [16,17]. Interestingly, they are very abundant in the EC lining of mammalian atheromas and in cultured EC that have been exposed to hypercholesterolemic serum [41]. A seminal paper showed the prompt formation of lipid droplets in intact murine aortic EC in vivo and ex vivo after a load of fatty acid [17]. This study suggests that beyond being an energy resource, endothelial lipid droplets represent a defense mechanism against lipotoxicity. Here, we show that 24 h culture in media containing high levels of glucose results in lipid droplet accumulation in HUVEC. To get insights into the involved mechanisms, we evaluated the amounts of PPARγ and its transcriptional coactivator EDF1. PPARγ is a ligand-activated transcription factor that is able to exert a broad spectrum of biological functions, including fatty acid handling and storage [42]. EDF1 is a low molecular weight protein that shuttles between the cytosol and the nucleus in response to environmental challenge [43] and is induced in HUVEC that have been exposed to oxidative stress [25]. When nuclear, it functions as a transcriptional coactivator for PPARγ [24,44]. HUVEC cultured in the presence of high glucose upregulate both PPARγ and EDF1. We hypothesize that the activation of the EDF1/PPARγ axis might fuel fatty acid synthesis in HUVEC. A similar conclusion was reached in HUVEC that had been cultured in a medium containing low levels of magnesium [25], thus suggesting that lipid accumulation is a common feature in EC exposed to metabolic stress. Moreover, culture in high glucose downregulates CPT1A, which consequently impairs lipid transport to the mitochondria. Accordingly, FAO is reduced in HUVEC that have been exposed to high amounts of glucose. We propose that the formation of lipid droplets in response to high amounts of glucose results from an imbalance between the synthesis and oxidation of fatty acids.

Whether lipid droplet-derived fatty acids are used as substrates for energy metabolism or for protection against lipoperoxidation in our experimental model remains to be elucidated. Other interesting aspects that we plan to investigate are the dynamics and the fate of these organelles. Moreover, a topic that we only mentioned briefly but that deserves more attention is the reason why SIRT1 appeared to be downregulated in our experimental setting. For this purpose, we recall that SIRT1 exerts beneficial effects on the endothelium, and consistently, antidiabetic drugs, anti-oxidants, and anti-inflammatory agents increase its amounts [45,46].

In conclusion, we identified TXNIP as one of the targets of VitD in HUVEC cultured in media containing high amounts of glucose. Therefore, VitD might represent a serviceable tool that can be used to control redox equilibrium with the aim of limiting or, at least, of delaying the onset of high glucose-induced endothelial dysfunction.

## Figures and Tables

**Figure 1 biomedicines-09-01874-f001:**
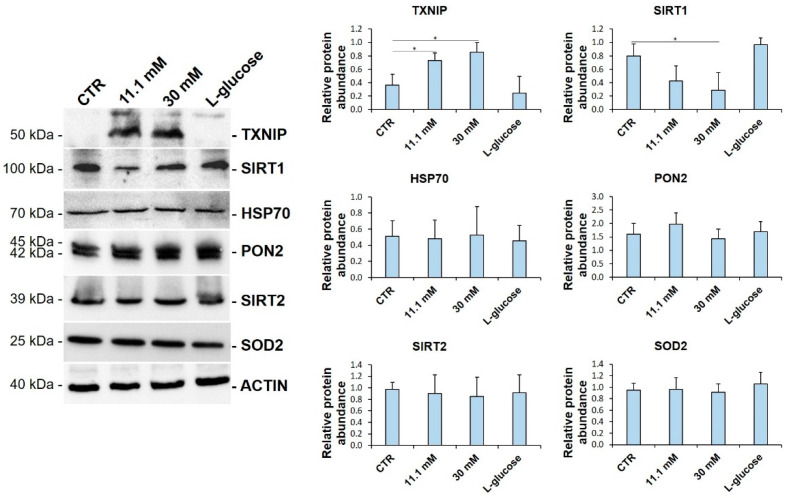
High glucose upregulates TXNIP and downregulates SIRT1 in HUVEC. Western blot (left panel) was performed on cell lysates using specific antibodies against TXNIP, SIRT1, HSP70, PON2, SIRT2, and SOD2. Actin was used as an equal loading control. A representative blot is shown. Densitometric analysis (right panel) was performed using Image J Lab software on three different blots, and the results are the mean of three independent experiments ± SD. * *p* < 0.05.

**Figure 2 biomedicines-09-01874-f002:**
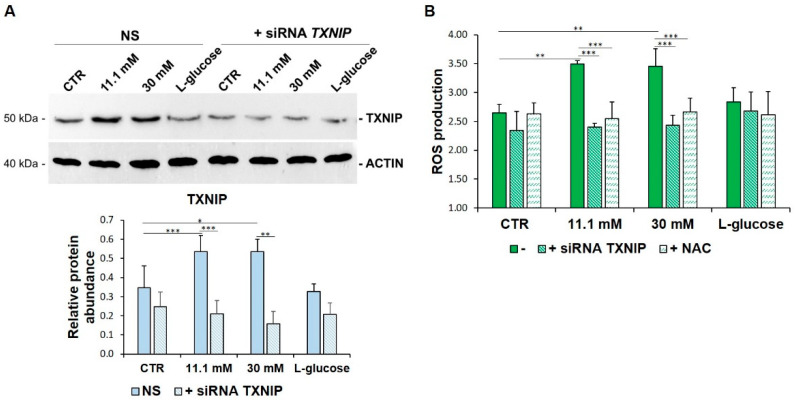
TXNIP upregulation is responsible for ROS accumulation in HUVEC cultured in high glucose levels. (**A**) HUVEC were cultured in medium containing 5 mM (CTR), 11.1 mM or 30 mM of d-glucose after *TXNIP* silencing. A scrambled non-silencing sequence (NS) was used as a control for silencing. Western blot (upper panel) was performed on cell lysates using specific antibodies against TXNIP. Actin was used an equal loading control. A representative blot is shown. Densitometric analysis (lower panel) was performed using Image J Lab software on three different blots, and the results are the mean of three independent experiments ± SD. (**B**) HUVEC were cultured in medium containing 5 mM (CTR), 11.1 mM or 30 mM of d-glucose (-) and either after *TXNIP* silencing or the addition of NAC 5 mM. l-glucose (30 mM) was used as a control of osmolarity. ROS production was evaluated by DCFH, as described in the methods. The results are the mean of three experiments performed in triplicate ± SD. * *p* < 0.05; ** *p* < 0.01; *** *p* < 0.001.

**Figure 3 biomedicines-09-01874-f003:**
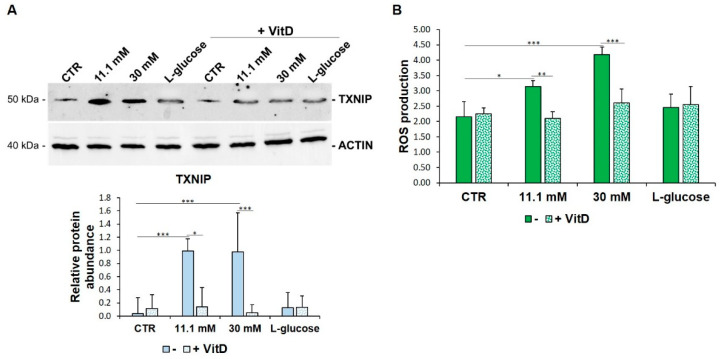
VitD prevents TXNIP upregulation and ROS accumulation in HUVEC cultured in high glucose conditions. (**A**) Western blot (upper panel) was performed on cell lysates using specific antibodies against TXNIP. Actin was used as an equal loading control. A representative blot is shown. Densitometric analysis (lower panel) was performed using Image J Lab software on three different blots, and the results are the mean of three independent experiments ± SD. (**B**) HUVEC were cultured in medium containing 5 mM (CTR), 11.1 mM, or 30 mM of d-glucose in the absence (-) or in the presence of VitD 20 nM. l-glucose (30 mM) was used as a control of osmolarity. ROS production was evaluated by DCFH, as described in the methods. The results are the mean of three experiments performed triplicate ± SD. * *p* < 0.05; ** *p* < 0.01; *** *p* < 0.001.

**Figure 4 biomedicines-09-01874-f004:**
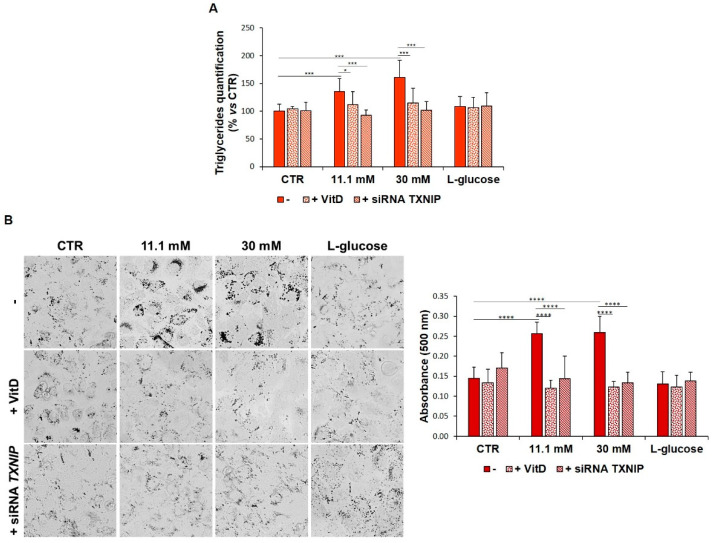
VitD and *TXNIP* silencing prevent high glucose-induced accumulation of triglycerides and lipid droplets. HUVEC were cultured in medium containing 5 mM (CTR), 11.1, or 30 mM of d-glucose (-) and either after *TXNIP* silencing or after the administration of 20 nM VitD. l-glucose (30 mM) was used as a control of osmolarity. (**A**) Triglyceride accumulation was measured by the Triglyceride Quantification Kit, as described in the methods. (**B**) The cells were stained with Oil Red O, and after the image acquisition using the FLoid Cell Imaging Station (Thermo Fisher Scientific) (left panel), the cells were solubilized, and the triglycerides were quantified by measuring the absorbance at 500 nm using the Varioskan LUX Multimode Microplate Reader (Thermo Fisher Scientific) (right panel). The results are the mean of three experiments that were repeated in triplicate ± SD. * *p* < 0.05; *** *p* < 0.001; **** *p* < 0.0001.

**Figure 5 biomedicines-09-01874-f005:**
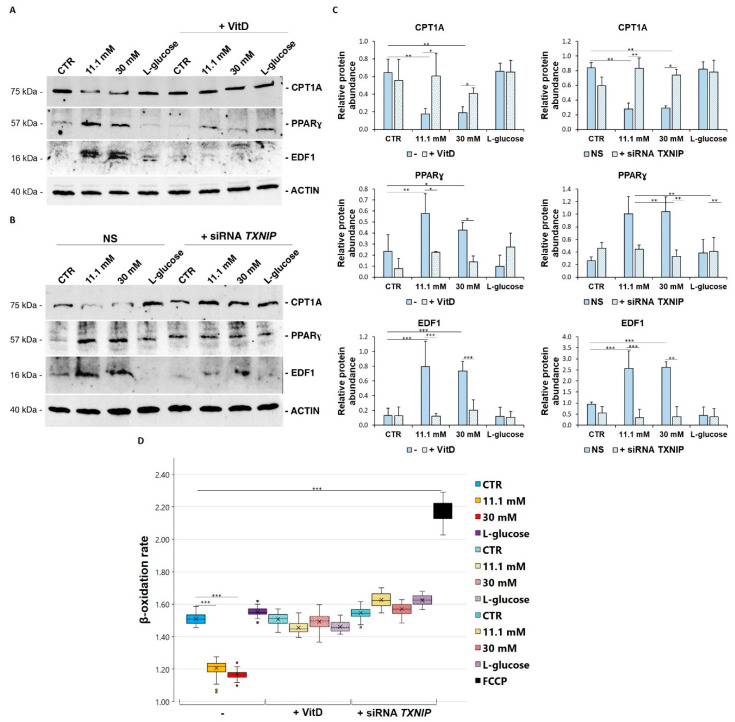
VitD and *TXNIP* silencing restore lipid metabolism in HUVEC cultured in high glucose. HUVEC were cultured in medium containing 5 mM (CTR), 11.1, or 30 mM of d-glucose (-) and either after *TXNIP* silencing or after the administration of 20 nM VitD. l-glucose (30 mM) was used as a control of osmolarity. (**A**,**B**) Western blot was performed on cell lysates using specific antibodies against CPT1A, PPARɣ, and EDF1. Actin was used as an equal loading control. A representative blot is shown. (**C**) Densitometric analysis was performed using Image J Lab software on three different blots, and the results are the mean of three independent experiments ± SD. (**D**) The β-oxidation rate was measured using a fatty acid oxidation assay kit, as described in the methods. The results are the mean of three experiments that were conducted in triplicate ± SD. * *p* < 0.05; ** *p* < 0.01; *** *p* < 0.001.

## Data Availability

Data are available in a publicly accessible repository. The data presented in this study are openly available in Dataverse at the following link: https://dataverse.unimi.it/dataverse/biomedicines/ accessed on 30 September 2021.

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
