# Peer review of "Vitamin D Prevents High Glucose-Induced Lipid Droplets Accumulation in Cultured Endothelial Cells: The Role of Thioredoxin Interacting Protein"

_biomedicines, 2021, doi:10.3390/biomedicines9121874_

Round 1

Reviewer 1 Report

The current study is investigating the protective effect of VitD on the vascular endothelial cells under hyperglycemic conditions. The authors mainly focused on unravelling the link between VitD3, TXNIP, oxidative stress, and lipid metabolism. Overall, the paper is well written and organized, and has high potential clinical impact.

Major comments:

The main conclusion and the tile of the study show that VitD prevents high glucose-induced lipid droplets accumulation by downregulating TXNIP. This is a strong conclusion. The experiemnts show that VitD reduced TXNIP which is associated with a decrease in lipid droplets. This doesn’t necessarilty mean that TXNIP is the only regulator for lipid droplet accumulation under hyperglycemic condition. Assessing lipid droplet accumulation in another group of cells treated with VitD and TXNIP mimetic or human TXNIP adenovirus will strengthen the study.

In the current study the authors showed the impact of hyperglycemia and VitD on TXNIP expression, but what about the thioredoxin activity? In the discussion the authors mentioned that TXNIP regulates lipid and glucose metabolism both dependently and independently from the inhibition of Thioredoxin (TRX).

The autors used 20nM VitD for all the experiements. It is not clear why they chose this specific concentration. Have the authors performed a dose response curve?

The western blot representative in figure 3A (TXNIP expression under 30 nM glucose) doesn’t match the densitometric analysis. Different representative is required.

A better western blot representative for EDF1 in Figure 5A is required. It is so hard to see the bands.

In the experimental design it is not clear if the cells were cultured in control/hyperglycemic conditions immediately after transfection.

Author Response

Dear reviewer, we uploaded the rebuttal as attachment. 

Reviewer 2 Report

The aim of the present study was to evaluate the effects of vitamin D in protecting Human Umbilical Vein Endothelial Cells (HUVEC) from oxidative stress, metabolic derangements, and lipid droplet accumulation. 

Abbreviation ("TXNIP") should not be used in the title, authors should use "Thioredoxin Interacting Protein".

Abbreviations should be defined at their appearance in the text ( for ex. TXNIP, ATP, NADPH, HUVEC, PFA)

"As end products of endothelial altered metabolism in high glucose, advanced glycation end products (AGE), which activate an inflammatory response, are generated" should be rephrased as it is not clear what the authors intended to express.

In the Results section, no references should be given nor comments on the results should be made. Some of the references should be moved to Material and Methods other to Discussion.

The discussion section is overall well written but limits of the present study and utility on the clinical practice of the present study should be presented here.

Author Response

(The authors gave the same response as above.)

Reviewer 3 Report

The authors present an interesting study in which they examine the potential role of TXNIP in the context of diabetes disease. Specifically, elevated levels of glucose have been demonstrated to a large degree to disturb the balance of redox levels within the vasculature; with persistent imbalance recognised as a key instigator of vascular dysfunction and consequently diabetes-driven vascular-related disease states. As such, there is a drive to understand the mechanisms related to such, and develop countermeasures and/or interventions towards reducing, or preferably, completely attenuating these effects driven by elevated blood glucose levels. The authors not only identify TXNIP as a prominent driver of the aforementioned physiological effects, but also highlight the therapeutic efficacy of Vitamin D supplementation of restoring redox balance through TXNIP pathways towards normal in an elevated glucose setting.

In reviewing the manuscript however, I had a number of concerns. The following should be addressed by the authors when preparing a suitable revision.

  1. The manuscript is well written, but there are instances where the language used, and the grammar, could be improved. For example, there are words that are not typical of scientific language used, and while the point is understandable, it is not fitting in this type of writing. The authors should review the writing of the entire manuscript in advance of any resubmission.
  2. The methods are of a good detail, however there are some aspects that could be improved. For example; for the Western blotting it would be useful if the specific antibody catalogue numbers were given and the respective concentrations of each individual antibody used to probe too.
  3. Were any non-transfected controls performed to examine the background effect of the conditions on the actual models?
  4. Details on the non-specific siRNA should be given.
  5. The authors do not allude to any measurements of viability per say. They do reference a previous article on cell number normalisation, however, it would be preferable if a) the authors provided data specific to this, and b) they gave albeit a brief description of how cell normalisation was performed in relation to these experiments.
  6. Why were HUVECs selected as the model of choice?
  7. How was the effective concentration of Vitamin D determined? Were any dose-dependent or time-dependent studies performed in relation to the effect of Vitamin D? For example, is continued presence of Vitamin D required to benefit from its effects?
  8. It would be useful if the authors could provide brightfield images of the cells in response to some of these conditions to validate their tolerance of the conditions. The images in Figure 4 are of too low contrast to determine how cell number may influence the results which are part of this article.
  9. The quality of the Westerns could be improved in some instances. There is a lot of non-specific noise in some blots, and some non-specific bands in other instance. How can the authors be sure that the results obtained represent the protein of interest? Were any controls blots ran using for example a recombinant form of the protein as a marker?
  10. The formatting of the graphs could be improved. The text in some instances is incredibly small, and difficult to read.
  11. In examining the knockdown using siRNA, the actual knockdown of TXNIP is not very impactful. Yes, the protein is reduced, but is its level important in the context of the cell? If this amount of knockdown can neutralise the damage exerted by 30mM of glucose, would more knockdown improve this for longer for example? Can the cells survive without this protein? More information on the optimisation of this knockdown would be appreciated.  
  12. In one set of experiments, using NAC was shown to more or less completely ablate the impact of elevated glucose on HUVECs. Are the authors suggesting that all elevated glucose derived ROS stem from NADPH oxidase? In this reviewers opinion there are several sources of ROS, so it interested me in this article that NAC alone could restore the redox balance.

Author Response

(The authors gave the same response as above.)

Round 2

Reviewer 3 Report

The authors have addressed a number of my comments and for that the manuscript is much improved and closer to publication standard. However, I do have some outstanding concerns:

  1. The antibody concentrations used are still not given.
  2. The labelling on many of the graphs remains quite small and hard to read.
  3. The authors suggest the BODIPY staining as a means of validating the cell number remains constant, however this stain is specific to cells expressing lipids which are present. The data does not provide any insight as to whether cells are lost in response to the conditions, and indeed whether they may impact on some of the readouts being provided.
  4. Regarding the concerns over cell presence/viability:
  5. More details on how the DAPI normalisation should be included
  6. More details on how the cells ‘were counted’ prior to triglyceride measurement should be given
  7. Are the authors suggesting MTT measurements were performed prior to the Oil Red O staining? Or was this performed on an independent set of cells whilst others were used for Oil Red O staining?
  8. The authors do not really address my concerns over the efficacy of NAC to completely ablate the levels of cellular ROS in this system.

Author Response

Dear Reviewer, please find attached our replies to your kind comments. Best regards. 

Round 3

Reviewer 3 Report

The authors have addressed many of my remaining concerns. I still think the formatting of the graphs could be improved. I acknowledge that the authors have made improvements to the labelling, but it is perhaps a case that the scale of the graphs be also adjusted to make them more legible. Many are improved, but in those where there are several pieces of data the details are still difficult to make out.